# Pollution and Health Risk Assessment of Hazardous Elements in Surface Dust along an Urbanization Gradient

Nargiza Kavsar [1], Mamattursun Eziz [1,2,*] and Nazupar Sidikjan [1]

1    College of Geographical Science and Tourism, Xinjiang Normal University, Urumqi 830054, China; nargizkavsar@126.com (N.K.); nazupar@126.com (N.S.)
2    Xinjiang Laboratory of Arid Zone Lake Environment and Resources, Xinjiang Normal University, Urumqi 830054, China
*    Correspondence: oasiseco@126.com; Tel./Fax: +86-991-4332295

**Abstract:** The hazardous element (HE) pollution of urban surface dust is a serious environmental issue, due to its toxicity and potential hazardous effects. Surface dust samples were collected from core urban, urban, and suburban gradients in the city of Urumqi in arid northwestern China, and the concentrations of six HEs, such as arsenic (As), cadmium (Cd), nickel (Ni), lead (Pb), mercury (Hg), and chromium (Cr) were determined. The pollution load index (*PLI*) and the US EPA health risk assessment model were applied, to assess and compare the pollution levels and the potential health risk of HEs in the surface dust in different urbanization gradients. The results indicated that the average concentrations of Hg, Cd, and Ni in the surface dust decreased in the order of core urban > urban > suburban, whereas the average concentrations of As, Cr, and Pb decreased in the order of urban > core urban > suburban. The *PLI* of HEs in surface dust decreased in the order of core urban > urban > suburban. The concentrations of HEs in the core urban and urban gradients were relatively higher than those in the suburban gradient. Furthermore, the total non-carcinogenic and carcinogenic risk index of the investigated HEs in surface dust decreased in the order of urban > core urban > suburban, for both adults and children. In addition, the pollution of surface dust by HEs in all urbanization gradients was more harmful to children's health than to adults'. Overall, the potential non-carcinogenic and carcinogenic health risk of the investigated HEs, instigated primarily via the oral ingestion of surface dust, was found to be within the acceptable range. However, urbanization has effected the accumulation of HEs in surface dust, and Cr was the main non-carcinogenic risk factor, whereas Cd was the main carcinogenic risk factor, among the analyzed HEs in surface dust in three urban gradients in the study area.

**Keywords:** urbanization gradient; surface dust; hazardous elements; pollution; health risk

## 1. Introduction

Surface dust is the "source" and "sink" of hazardous elements (HEs) in urban environments, and is closely related to the urban ecosystem and to human health [1,2]. The concentrations of HEs in urban surface dust have become one of the crucial eco-environmental problems in urban settings [3,4]. The increasing demand for metals in industries and urbanization processes can strongly disturb the natural geochemical cycling of HEs in urban ecosystem [5,6], and can result in the accumulation of HEs in surface dust in an urban environment [7,8]. Traffic exhaust fumes, incinerators, industrial waste, and the atmospheric deposition of dust and aerosols have continuously added HEs to the urban environment [9]. However, the accumulation of HEs in the human body can cause irreversible damage to human health [10,11]. Therefore, HEs in surface dust can serve as a representative indicator of the urban environmental quality [12].

The pollution of urban surface dust by HEs, whether through natural or anthropogenic sources, is a growing environmental problem, due to their potential toxicity, and their

hazardous effects on the urban eco-environment and human health [2,13,14]. Hazardous elements in surface dust can transmit from the ground surface to soil and water, and can easily enter the human body through direct touch, breathing system inhalation, and hand-to-mouth intake [15,16]. Exposure to HEs has been known to cause serious systemic health issues, such as damage to the kidneys and liver, breast and gastrointestinal cancer, respiratory diseases, neurological disorders, anemia, skin lesions, renal diseases, and congenital malformation [17,18]. Therefore, assessment of the pollution from, and potential health risk of, HEs in urban surface dust has emerged as a new topic at the forefront of environmental research.

Recently, many studies have analyzed the pollution risk of HEs in soil along an urbanization gradient. Celine et al. [19] reported that there is a distinctly different association among the HEs in park soils in urban and suburban areas, and in the countryside, in Hong Kong. Lu et al. [20] analyzed the HEs in soil along a typical urban–rural gradient zone in a quickly growing city in eastern China, and found that soils in the urban gradient zone were relatively more highly concentrated with Cd, Cu, Pb, and Zn. Some other studies [21–24] also reported that the urbanization process could affect not only the concentrations, but also the spatial distribution patterns, of HEs in soil. Streeter et al. [25] suggested that the degree of regional urbanization and industrialization could sufficiently differentiate urban soils from surrounding rural agricultural soils. Hong et al. suggested that the pollution level of HEs in urban road-deposited sediment was usually higher than that in a rural gradient zone [10]. Furthermore, urbanization can affect the accumulation of HEs in soil in the arid zone cities [26].

The above studies mainly involved the HEs in soil along various urbanization gradient zones. During this time, quite a few studies [27,28] have considered the HEs in indoor environments. However, there are very few studies focused on the heavy metal pollution of urban surface dust along an urbanization gradient. One recent study reported that urbanization can significantly affect the pollution of soil by HEs along an urban–rural gradient zone in Urumqi [26]. So far, however, there has been no corresponding discussion about the pollution of surface dust by HEs along an urbanization gradient. Obviously, it is very necessary to understand the potential health impacts of HEs in surface dust along an urban gradient zone.

In view of the shortage of current research, surface dust samples from a typical urbanization gradient zone in the city of Urumqi in China were collected, and the concentrations of six HEs were measured. The main objectives of this research work were to identify the pollution levels of HEs in surface dust along an urbanization gradient, and to compare the potential health risks of HEs on adults and children via oral ingestion, inhalation, and dermal contact. The results of this study will provide theoretical and technical support in the protection of human health and the eco-environmental safety of cities in the arid zone.

## 2. Materials and Methods

### 2.1. Study Area

The city of Urumqi is one of the most important cities in the "Silk Road Economic Belt", Xinjiang, in the northwestern arid zone of China. This city is situated in the middle part of the Tianshan Mountains, and lies within the geographical coordinates of 87°28′–87°45′ E and 43°42′–43°54′ N, with a total built-up area of more than 500 km$^2$. The climate of this city is a typical continental desert climate [26,29]. A continuously distributed typical "core-urban–urban–suburban" gradient zone, as illustrated in Figure 1, was chosen in the city of Urumqi to identify the effects of urbanization on the concentrations of HEs in urban surface dust. The core urban (the area within the first ring road of Urumqi, with a high density of buildings and roads), urban (the area between the first and second ring roads of Urumqi, with a higher density of buildings and roads), and suburban (areas between the second ring road and the rural areas) gradients were divided according to previous research [26].

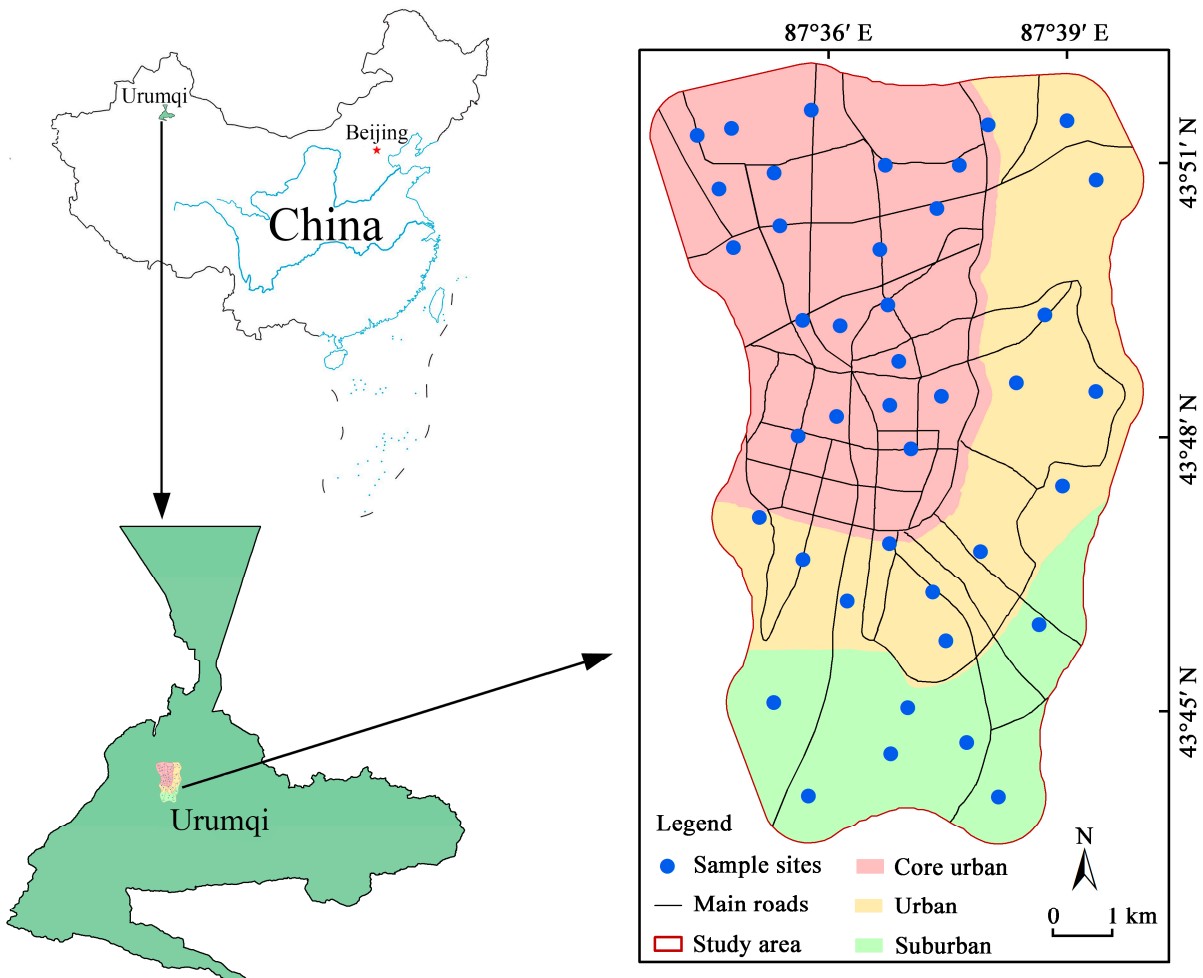

**Figure 1.** Locations of the city of Urumqi, and the sample sites.

### 2.2. Sample Collection, Preparation, and Analysis

A total of 41 surface dust samples were collected from the core urban, urban, and suburban gradients in the city of Urumqi [26]. The sample sites are shown in Figure 1. Considering the spatial heterogeneity of HEs in surface dust in different urban gradients [10,15,18], 21, 13, and 7 dust samples were collected from the core urban, urban, and suburban gradients, respectively. The sample collection methods adopted in this research work were those introduced in "NY/T 395–2000" [30]. At each sample site, about 10 subsamples of surface dust were collected from the road surface, using a polyethylene brush, and the subsamples were mixed as one composite surface dust sample, and sent to the laboratory.

As described in "HJ/T 166–2004" [31], all surface dust samples were air-dried, ground, and then sieved through a 0.15 mm nylon mesh, and digested. Finally, atomic fluorescence spectrometry (PERSEE, PF–7, Fuzhou, China) was used to determine the As in the surface dust samples, while a flame–flameless atomic absorption spectrophotometer (Agilent 200AA, Santa Clara, CA, USA) was used to measure the Cd, Ni, Pb, Hg, and Cr concentrations in the collected surface dust samples. The analytical data quality was assessed based on laboratory quality control methods, such as reagent blanks and duplicates. A standard solution of analyzed HEs was used to compare the collected surface dust samples with the national standards samples (GSS-12), for the accuracy of the analytical procedures. The recovery rate of the surface dust samples mixed with standard samples ranged from 93.67% to 105.86%.

### 2.3. Pollution Assessment of HEs

The pollution status of the HEs in surface dust was evaluated using the pollution load index (*PLI*) [32] to directly reflect the degree of enrichment by HEs of the surface dust [18]. The calculation formula for the *PLI* value of HEs is as follows:

$$CF_i = C_i/C_b \tag{1}$$

$$PLI = \sqrt[n]{CF_1 \times CF_2 \times \cdots CF_n} \tag{2}$$

where $CF_i$ represents the contamination factor of a single HE *i*, and $C_i$ and $C_b$ represent the measured and background concentration [18] of a single HE *i*, respectively. The following criteria were used to classify the pollution grades of the *CF* and *PLI*, respectively: no pollution ($CF \leq 0.7$), slight ($0.7 < CF \leq 1$), low ($1 < CF \leq 2$), moderate ($2 < CF \leq 3$), and heavy ($CF \geq 3$) pollution; and no pollution ($PLI \leq 1$), low ($1 < PLI \leq 2$), moderate ($2 < PLI \leq 3$), and heavy ($PLI \geq 3$) pollution [18,26].

### 2.4. Health Risk Assessment of HEs

The US EPA health risk assessment model, which includes both non-carcinogenic and carcinogenic risk assessments, has been considered a useful tool for detecting the potential health risks of HEs in surface dust [1,3,8].

#### 2.4.1. Exposure Analysis

The exposure levels of HEs in surface dust were evaluated on the basis of the chronic daily intake (*CDI*, mg/kg/day). The calculation formula for the *CDI* values for the oral ingestion, inhalation, and dermal contact exposure routes were as follows [33–35]:

$$CDI_{\text{ingest}} = (C_i \times IngR \times CF \times EF \times ED)/(BW \times AT) \tag{3}$$

$$CDI_{\text{inhale}} = (C_i \times InhR \times EF \times ED)/(PEF \times BW \times AT) \tag{4}$$

$$CDI_{\text{dermal}} = (C_i \times SA \times AF \times ABS \times EF \times ED)/(BW \times AT) \tag{5}$$

$$CDI_{\text{total}} = CDI_{\text{ingest}} + CDI_{\text{inhale}} + CDI_{\text{dermal}} \tag{6}$$

The exposure parameters for *CDI* estimation and their meanings are given in Table 1.

**Table 1.** The *CDI* estimation parameters.

| Parameters | Meaning and Units | Children | Adult |
|:---:|:---:|:---:|:---:|
| *IngR* | Consumption rate of dusts (mg/d) | 200 | 100 |
| *InhR* | Dust inhalation rate (m$^3$/d) | 7.5 | 16.2 |
| *CF* | Unit conversion factor (kg/mg) | $1 \times 10^{-6}$ | $1 \times 10^{-6}$ |
| *EF* | Exposure frequency (d/a) | 350 | 350 |
| *ED* | Exposure duration (year) | 6 | 30 |
| *SA* | Exposed skin area (cm$^2$) | 899 | 1600 |
| *AF* | Skin adherence factor (mg/(cm$^2$/d)) | 0.20 | 0.07 |
| *PEF* | Particulate emission factor (m$^3$/kg) | $1.36 \times 10^9$ | $1.36 \times 10^9$ |
| *BW* | Average body weight (kg) | 21.2 | 62.4 |
| $AT_{nc}$ | Average exposure time for non-cancer (d) | $365 \times ED$ | $365 \times ED$ |
| $AT_{ca}$ | Average exposure time for cancer (d) | $365 \times 70$ | $365 \times 70$ |
| *ABS* | Dermal absorption factor (unitless) | Hg = Cr = Ni = Pb = 0.01; As = 0.03; Cd = 0.005 | |

2.4.2. Non-Carcinogenic Risk Assessment

The hazard index (*HI*), which is based on the hazard quotient (*HQ*) of a single HE, was used to quantify the non-carcinogenic health risk of HEs. The *HQ* and *HI* can be estimated as follows:

$$HQ = \frac{CDI}{RfD} \tag{7}$$

$$HI = \sum HQ = HQ_{\text{ingest}} + HQ_{\text{inhale}} + HQ_{\text{dermal}} \tag{8}$$

where *RfD* represents the reference dose (mg/kg/day), which is regarded as an estimation of daily exposure to the human population. A *HQ* or *HI* < 1 denotes a negligible non-carcinogenic health risk of HEs, while a *HQ* or *HI* ≥ 1 means that the HEs in the surface dust may pose a potential non-carcinogenic risk [36].

2.4.3. Carcinogenic Risk Assessment

The carcinogenic risks of As, Cd, Cr, and Ni, which are regarded as carcinogenic HEs [26], were calculated using Equations (9) and (10).

$$CR = CDI \times SF \tag{9}$$

$$TCR = \sum CR = CR_{\text{ingest}} + CR_{\text{inhale}} + CR_{\text{dermal}} \tag{10}$$

where *CR* indicates the carcinogenic risk of a single HE (unitless), *TCR* represents the total carcinogenic risk posed by all HEs in surface dust (unitless), and *SF* represents the carcinogenic slope factor (mg/kg/day). A *CR* or *TCR* < $10^{-6}$ denotes a negligible carcinogenic health risk in HEs, while an *CR* or *TCR* ≥ $10^{-4}$ means that the HEs in surface dust may cause a potential carcinogenic risk, and if $10^{-6} \leq CR$ or $TCR \leq 10^{-4}$, it means that the potential carcinogenic risk posed by the HEs in surface dust is acceptable or tolerated [37]. The *RfD* and *SF* values of the HEs in surface dust were defined according to the relevant research results [38,39], as given in Table 2.

**Table 2.** The *RfD* for non-carcinogenic elements, and *SF* for carcinogenic elements.

| Elements | *RfD*/(mg/kg/d) | | | *SF*/(mg/kg/d) | | |
|:---:|:---:|:---:|:---:|:---:|:---:|:---:|
| | Ingestion | Inhalation | Dermal | Ingestion | Inhalation | Dermal |
| Pb | 0.0035 | 0.00352 | 0.000525 | / | / | / |
| Ni | 0.020 | 0.0206 | 0.0054 | / | 0.84 | / |
| As | 0.0003 | 0.000123 | 0.0003 | 1.50 | 0.0043 | 1.50 |
| Cd | 0.001 | 0.001 | 0.00001 | / | 6.30 | / |
| Hg | 0.0003 | 0.0003 | 0.000024 | / | / | / |
| Cr | 0.003 | 0.0000286 | 0.00006 | 0.50 | 42.0 | / |

## 3. Results and Discussion

*3.1. Concentration of HEs in Surface Dust along the Urbanization Gradient*

As given in Table 3, on average, the concentrations of As, Hg, Cd, Cr, Ni, and Pb in the surface dusts in the core urban gradient were 9.14 mg/kg, 0.18 mg/kg, 0.24 mg/kg, 63.83 mg/kg, 36.95 mg/kg, and 36.61 mg/kg, respectively. The average concentrations of these six HEs in the surface dusts in the urban gradient were 9.96 mg/kg, 0.14 mg/kg, 0.21 mg/kg, 65.52 mg/kg, 32.99 mg/kg, and 40.28 mg/kg, respectively. Finally, the average concentrations of these six HEs in the suburban gradient were 8.61 mg/kg, 0.13 mg/kg, 0.19 mg/kg, 61.13 mg/kg, 31.39 mg/kg, and 27.11 mg/kg, respectively. The average concentrations of the Hg, Cr, Ni, and Pb elements in the surface dust in all urbanization gradients, and of Cd in the core urban surface dust, exceeded the corresponding background values, indicating the highest enrichment level of mercury in the surface dust in all the urbanization gradients in the study area.

**Table 3.** Concentrations of HEs in surface dust along the urbanization gradient.

| Gradient | Statistics | As | Hg | Cd | Cr | Ni | Pb |
|---|---|---|---|---|---|---|---|
| Core urban (*n* = 21) | Minimum/(mg/kg) | 5.30 | 0.07 | 0.09 | 50.07 | 21.61 | 16.00 |
| | Maximum/(mg/kg) | 14.20 | 0.55 | 0.50 | 81.08 | 74.94 | 56.30 |
| | Average/(mg/kg) | 9.14 | 0.18 | 0.24 | 63.83 | 36.95 | 36.61 |
| | St.D/(mg/kg) | 2.42 | 0.11 | 0.12 | 7.91 | 13.22 | 11.09 |
| | CV | 0.26 | 0.61 | 0.50 | 0.12 | 0.36 | 0.30 |
| Urban (*n* = 13) | Minimum/(mg/kg) | 5.00 | 0.07 | 0.12 | 45.01 | 27.34 | 18.80 |
| | Maximum/(mg/kg) | 15.90 | 0.29 | 0.36 | 94.38 | 47.43 | 146.00 |
| | Average/(mg/kg) | 9.96 | 0.14 | 0.21 | 65.52 | 32.99 | 40.28 |
| | St.D/(mg/kg) | 3.05 | 0.07 | 0.06 | 12.66 | 6.58 | 31.23 |
| | CV | 0.31 | 0.50 | 0.29 | 0.19 | 0.20 | 0.78 |
| Suburban (*n* = 7) | Minimum/(mg/kg) | 8.00 | 0.07 | 0.09 | 48.01 | 18.20 | 19.20 |
| | Maximum/(mg/kg) | 9.60 | 0.25 | 0.35 | 74.97 | 39.04 | 44.00 |
| | Average/(mg/kg) | 8.61 | 0.13 | 0.19 | 61.13 | 31.39 | 27.11 |
| | St.D/(mg/kg) | 0.48 | 0.06 | 0.09 | 9.10 | 6.73 | 7.59 |
| | CV | 0.06 | 0.46 | 0.47 | 0.15 | 0.21 | 0.28 |
| Background value * | | 9.99 | 0.076 | 0.23 | 53.20 | 29.90 | 14.10 |

Note: * Background values refer to the concentrations of HEs in the soil in Urumqi.

Obviously, the decreasing order of the average concentrations of Hg, Cd, and Ni in the surface dust was: core urban > urban > suburban, whereas the decreasing order of the average concentrations of As, Cr, and Pb in the surface dust was: urban > core urban > suburban. This suggests that the concentrations of HEs in surface dust differ among the investigated urbanization gradients, and that the surface dust was less enriched with HEs in the suburban gradient, in comparison with the core urban and urban gradients, as HEs are mainly originated by fuel combustion, and are emitted into the atmosphere with exhaust gases [40]. Cd exists in lubricating oil, tires, and brakes, and comes mainly from traffic sources [41]. HEs such as Cr and Ni can enter the atmosphere together with industrial waste gases. The major origins of lead are traffic exhaust fumes and fuel combustion. Industrial activities such as small-scale gold mining and non-ferrous metal production may be the main origins of mercury [40–42]. Additionally, the Hg in urban surface dust may originate from commercial activities and family sources in urban areas [18]. The HEs released from the soil then enter the atmosphere, and can re-enter the surface dust through sedimentation, and may re-suspend as particulate matter.

According to the classification criteria of the coefficient of variations (CV) and the obtained CVs of the HEs in the surface dust in each urbanization gradient, the Hg in all the urbanization gradients, the Cd in the core urban and suburban gradients, and the Pb in the urban gradient were highly variable (CV > 36%), showing that these HEs in the corresponding urbanization gradients vary significantly across the sample sites, and that the possible origins of these HEs are most likely influenced by anthropogenic activities. Meanwhile, the As in the core urban and urban gradients, the Cd and Cr in the urban gradient, the Ni in all gradients, and the Pb in the core urban and suburban gradients were moderately variable (16% < CV ≤ 36%), indicating that these HEs are most likely influenced by both anthropogenic and natural factors. However, the As in the suburban gradient and the Cr in the core urban and suburban gradients showed a low variability (CV < 16%), indicating that these two HEs in the corresponding urbanization gradients are controlled by natural factors.

### 3.2. Spatial Distribution of Concentration of HEs in Surface Dust

The spatial distribution of the concentrations of the investigated HEs in the surface dust in the city of Urumqi was mapped using a GIS-based ordinary Kriging (OK) interpolation method (Figure 2). The spatial distributions of As and Pb in the surface dust, as illustrated in Figure 2, were similar to each other, with higher concentrations of these two

HEs seen primarily in the core urban and urban gradients, and low concentrations seen mainly in the suburban gradient. This findings are in agreement with the conclusion of a related study [43].

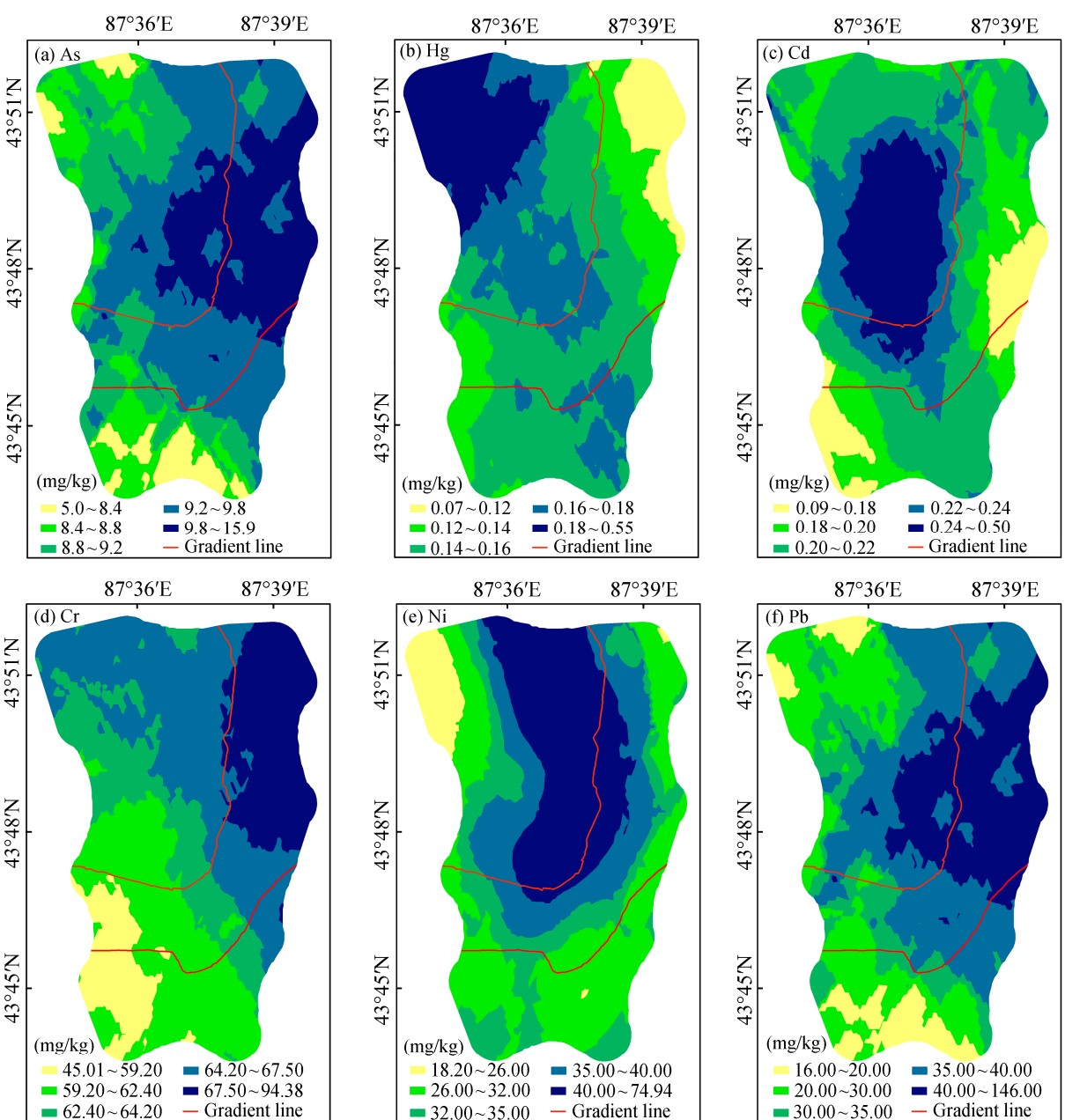

**Figure 2.** Spatial distribution of the concentrations of HEs in the surface dust.

A zonal spatial distribution pattern for the HEs, including Hg, Cd, and Ni was found in this study, with the highest accumulation observed in the core urban gradient, and a lower accumulation observed in the suburban gradient. These three HE concentrations decreased from the core urban gradient to the suburban gradient in the study area. In the case of Cr, a zonal spatial distribution pattern was also observed in this study, with higher concentrations observed in the urban gradient, and lower concentrations observed in the suburban gradient. The Cr concentrations decreased between the northeastern parts and the southwestern parts of the study area. However, low concentrations of all the HEs in this study were seen in the suburban gradient, with a low road density, traffic flow, population density, and industrial production. Overall, the concentrations of the HEs in

the surface dust in the core urban and the urban gradients were relatively higher than in the suburban gradient, which seems to be a clear indication that urbanization can influence the accumulation of HEs in the surface dust in the study area.

### 3.3. Pollution of HEs in Surface Dust along the Urbanization Gradient

As given in Table 4, the descending order of the pollution levels of the HEs in the surface dust varied under different urban gradients. On average, the *CF* values of the analyzed HEs in the surface dust in the core urban gradient could be ranked: Pb(2.60) > Hg(2.34) > Ni(1.24) > Cr(1.20) > Cd(1.02) > As(0.91), while the *CF* values of the HEs in the urban surface dust could be ranked: Pb(2.86) > Hg(1.89) > Cr(1.23) > Ni(1.10) > As(1.00) > Cd(0.90), and the *CF* values of the HEs in the suburban surface dust could be ranked: Pb(1.92) > Hg(1.73) > Cr(1.15) > Ni(1.05) > As(0.86) > Cd(0.80).

**Table 4.** Pollution levels of HEs in surface dust along the urbanization gradient.

| Gradient | Statistics | CF | | | | | | PLI |
|---|---|---|---|---|---|---|---|---|
| | | As | Hg | Cd | Cr | Ni | Pb | |
| Core urban (*n* = 21) | Minimum | 0.53 | 0.95 | 0.41 | 0.94 | 0.72 | 1.13 | 0.94 |
| | Maximum | 1.42 | 7.24 | 2.17 | 1.52 | 2.51 | 3.99 | 1.97 |
| | Average | 0.91 | 2.34 | 1.02 | 1.20 | 1.24 | 2.60 | 1.35 |
| Urban (*n* = 13) | Minimum | 0.50 | 0.93 | 0.52 | 0.85 | 0.91 | 1.33 | 1.04 |
| | Maximum | 1.59 | 3.82 | 1.57 | 1.77 | 1.59 | 10.35 | 1.61 |
| | Average | 1.00 | 1.89 | 0.90 | 1.23 | 1.10 | 2.86 | 1.29 |
| Suburban (*n* = 7) | Minimum | 0.80 | 0.97 | 0.37 | 0.90 | 0.61 | 1.36 | 0.87 |
| | Maximum | 0.96 | 3.29 | 1.52 | 1.41 | 1.31 | 3.12 | 1.56 |
| | Average | 0.86 | 1.73 | 0.80 | 1.15 | 1.05 | 1.92 | 1.15 |

According to the grading criteria and the calculated values of *CF*, the surface dust was polluted at a low level by Cr and Ni, while slightly polluted by As in all the urbanization gradients; a moderate pollution by Hg was observed in the core urban gradient, while a moderate pollution by Pb was observed in the core urban and urban gradients. Additionally, the urban and suburban surface dusts were polluted at a low level by Hg, and slightly polluted by Cd. Meanwhile, the Cd in the core urban gradient and the Pb in the suburban gradient showed a low pollution level.

However, the average *CF* values of Hg, Cd, and Ni in the surface dust decreased in the order: core urban > urban > suburban, while the average *CF* values of As, Cr and Pb in the surface dust decreased in the order: urban > core urban > suburban. The results indicate that the surface dusts in the suburban gradient, with a relatively low population density and traffic flow, are relatively clean in comparison with the surface dusts in the core urban and urban gradients. Overall, hazardous elements, particularly Hg and Pb, are likely to be a significant pollutant of surface dust in all the urbanization gradients in the city of Urumqi. Therefore, they should be closely monitored.

The average *PLI* values of the HEs in the surface dust in the core urban, urban, and suburban gradients were 1.35, 1.29, and 1.15, respectively, at the low pollution level. The *PLI* of the HEs can be ranked: core urban > urban > suburban. The average *PLI* values of the HEs in the surface dust in the core urban gradient were 4.65% and 17.39% higher than those in the urban and the suburban gradients, respectively. Overall, in all the urban gradients, mercury contributed the most to the *PLI* of the HEs in surface dust, accounting for 57.69%, 68.25%, and 66.47% of the *PLI* of the HEs in the surface dust in the core urban, urban, and suburban gradients, respectively. This indicates that Hg is the most significant pollution factor in the surface dust in all the urban gradient zones in the city of Urumqi.

### 3.4. Non-Carcinogenic Risk of HEs in Surface Dust along the Urbanization Gradient

The hazard quotients (*HQs*) of each HE in the surface dust in all the urbanization gradients via three exposure routes were estimated for adults and children, and then the cumulative effect of the *HQs* of the analyzed HEs was estimated, using the hazard indexes (*HIs*). The potential health risks of the HEs in the different urbanization gradients were compared and discussed.

As given in Table 5, for both adults and children, the average *HQ* values of the investigated HEs in the surface dust in the core urban, urban, and suburban gradients followed the order: $HQ_{Cr} > HQ_{As} > HQ_{Pb} > HQ_{Ni} > HQ_{Cd} > HQ_{Hg}$. For children, the *HQ* values of Cr were higher than those of the other HEs, and they accounted for 52.53%, 51.26%, and 54.59% of the corresponding *HI* values of the surface dust in the core urban, urban, and suburban gradients, respectively, compared to 55.77%, 54.46%, and 57.74% of the *HIs* for adults, respectively. These results imply that, across all the urban gradient zones, Cr contributed the most to the total *HI* value of the investigated HEs in surface dust, and Cr is the main non-carcinogenic risk factor in the surface dust, with the highest potential non-carcinogenic health risk.

**Table 5.** Non-carcinogenic risk index of the HEs in surface dust.

| Gradient | Metal | $HQ_{ingest}$ | | $HQ_{inhale}$ | | $HQ_{dermal}$ | | $HQ$ | | $HI$ | |
|---|---|---|---|---|---|---|---|---|---|---|---|
| | | Children | Adults | Children | Adults | Children | Adults | Children | Adults | Children | Adults |
| Core urban | As | $2.91 \times 10^{-1}$ | $4.68 \times 10^{-2}$ | $1.96 \times 10^{-5}$ | $1.36 \times 10^{-5}$ | $2.62 \times 10^{-3}$ | $5.24 \times 10^{-4}$ | $2.94 \times 10^{-1}$ | $4.73 \times 10^{-2}$ | | |
| | Hg | $5.66 \times 10^{-3}$ | $9.10 \times 10^{-4}$ | $1.56 \times 10^{-7}$ | $1.08 \times 10^{-7}$ | $6.36 \times 10^{-4}$ | $1.27 \times 10^{-4}$ | $6.29 \times 10^{-3}$ | $1.04 \times 10^{-3}$ | | |
| | Cd | $2.25 \times 10^{-3}$ | $3.62 \times 10^{-4}$ | $6.20 \times 10^{-8}$ | $4.31 \times 10^{-8}$ | $6.06 \times 10^{-3}$ | $1.22 \times 10^{-3}$ | $8.31 \times 10^{-3}$ | $1.58 \times 10^{-3}$ | 0.910 | 0.158 |
| | Cr | $2.03 \times 10^{-1}$ | $3.27 \times 10^{-2}$ | $5.88 \times 10^{-4}$ | $4.09 \times 10^{-4}$ | $2.74 \times 10^{-1}$ | $5.49 \times 10^{-2}$ | $4.78 \times 10^{-1}$ | $8.80 \times 10^{-2}$ | | |
| | Ni | $1.76 \times 10^{-2}$ | $2.84 \times 10^{-3}$ | $4.72 \times 10^{-7}$ | $3.28 \times 10^{-7}$ | $5.88 \times 10^{-4}$ | $1.18 \times 10^{-4}$ | $1.82 \times 10^{-2}$ | $2.96 \times 10^{-3}$ | | |
| | Pb | $9.99 \times 10^{-2}$ | $1.61 \times 10^{-2}$ | $2.74 \times 10^{-6}$ | $1.90 \times 10^{-6}$ | $5.99 \times 10^{-3}$ | $1.20 \times 10^{-3}$ | $1.06 \times 10^{-1}$ | $1.73 \times 10^{-2}$ | | |
| Urban | As | $3.17 \times 10^{-1}$ | $5.10 \times 10^{-2}$ | $2.13 \times 10^{-5}$ | $1.48 \times 10^{-5}$ | $2.85 \times 10^{-3}$ | $5.71 \times 10^{-4}$ | $3.20 \times 10^{-1}$ | $5.16 \times 10^{-2}$ | | |
| | Hg | $4.58 \times 10^{-3}$ | $7.37 \times 10^{-4}$ | $1.26 \times 10^{-7}$ | $8.78 \times 10^{-8}$ | $5.15 \times 10^{-4}$ | $1.03 \times 10^{-4}$ | $5.09 \times 10^{-3}$ | $8.40 \times 10^{-4}$ | | |
| | Cd | $1.98 \times 10^{-3}$ | $3.36 \times 10^{-4}$ | $5.47 \times 10^{-8}$ | $3.80 \times 10^{-8}$ | $5.35 \times 10^{-3}$ | $1.07 \times 10^{-3}$ | $7.33 \times 10^{-3}$ | $1.39 \times 10^{-3}$ | 0.956 | 0.166 |
| | Cr | $2.09 \times 10^{-1}$ | $3.36 \times 10^{-2}$ | $6.03 \times 10^{-4}$ | $4.19 \times 10^{-4}$ | $2.81 \times 10^{-1}$ | $5.64 \times 10^{-2}$ | $4.90 \times 10^{-1}$ | $9.04 \times 10^{-2}$ | | |
| | Ni | $1.58 \times 10^{-2}$ | $2.53 \times 10^{-3}$ | $4.22 \times 10^{-7}$ | $2.93 \times 10^{-7}$ | $5.25 \times 10^{-4}$ | $1.05 \times 10^{-4}$ | $1.63 \times 10^{-2}$ | $2.64 \times 10^{-3}$ | | |
| | Pb | $1.10 \times 10^{-1}$ | $1.77 \times 10^{-2}$ | $3.01 \times 10^{-6}$ | $2.09 \times 10^{-6}$ | $6.59 \times 10^{-3}$ | $1.32 \times 10^{-3}$ | $1.17 \times 10^{-1}$ | $1.90 \times 10^{-2}$ | | |
| Suburban | As | $2.74 \times 10^{-1}$ | $4.41 \times 10^{-2}$ | $1.84 \times 10^{-5}$ | $1.28 \times 10^{-5}$ | $2.47 \times 10^{-3}$ | $4.94 \times 10^{-4}$ | $2.77 \times 10^{-1}$ | $4.46 \times 10^{-2}$ | | |
| | Hg | $4.18 \times 10^{-3}$ | $6.72 \times 10^{-4}$ | $1.15 \times 10^{-7}$ | $8.00 \times 10^{-8}$ | $4.69 \times 10^{-4}$ | $9.40 \times 10^{-5}$ | $4.64 \times 10^{-3}$ | $7.66 \times 10^{-4}$ | | |
| | Cd | $1.77 \times 10^{-3}$ | $2.84 \times 10^{-4}$ | $4.87 \times 10^{-8}$ | $3.39 \times 10^{-8}$ | $4.77 \times 10^{-3}$ | $9.55 \times 10^{-4}$ | $6.53 \times 10^{-3}$ | $1.24 \times 10^{-3}$ | 0.839 | 0.146 |
| | Cr | $1.95 \times 10^{-1}$ | $3.13 \times 10^{-2}$ | $5.63 \times 10^{-4}$ | $3.91 \times 10^{-4}$ | $2.62 \times 10^{-1}$ | $5.26 \times 10^{-2}$ | $4.58 \times 10^{-1}$ | $8.43 \times 10^{-2}$ | | |
| | Ni | $1.50 \times 10^{-2}$ | $2.41 \times 10^{-3}$ | $4.01 \times 10^{-7}$ | $2.79 \times 10^{-7}$ | $4.99 \times 10^{-4}$ | $1.00 \times 10^{-4}$ | $1.55 \times 10^{-2}$ | $2.51 \times 10^{-3}$ | | |
| | Pb | $7.40 \times 10^{-2}$ | $1.19 \times 10^{-2}$ | $2.03 \times 10^{-6}$ | $1.41 \times 10^{-6}$ | $4.43 \times 10^{-3}$ | $8.89 \times 10^{-4}$ | $7.84 \times 10^{-2}$ | $1.28 \times 10^{-2}$ | | |

As for the exposure routes, the average *HQ* values of the investigated HEs in the surface dust in all the urban gradient zones followed the order: $HQ_{ingest} > HQ_{dermal} > HQ_{inhale}$. This implies that unconscious ingestion was the primary route of exposure to the potential non-carcinogenic risks of the HEs in the surface dust in the city of Urumqi.

The *HI* values of the HEs in the surface dust in the core urban, urban, and suburban gradients were 0.910, 0.956, and 0.839 for children, respectively, compared to 0.158, 0.166, and 0.146 for adults, respectively. The obtained *HIs* of the HEs in surface dust for children were much higher than those for adults. This indicates that the HEs in surface dust may pose higher non-carcinogenic risks to children than to adults. This finding can be interpreted through the possibility that children's hemoglobin has a relatively higher sensitivity to the HEs in surface dust, and they can absorb HEs relatively faster than adults [6,44].

On the whole, according to the classification criteria for a non-carcinogenic health risk, the *HQ* and *HI* values of the investigated HEs in the surface dust in all the urban gradients were lower than 1, for both children and adults. This suggests that the non-carcinogenic health risk of the HEs to humans can be negligible. Moreover, the obtained *HI* values of the HEs for adults and children could be ranked: $HI_{urban} > HI_{core\ urban} > HI_{suburban}$, indicating that the HEs in the surface dust in the suburban gradient had a lower potential health risk than those in the other urban gradients in the study area.

### 3.5. Carcinogenic Risk of HEs in Surface Dust along the Urbanization Gradient

The As, Cd, Cr, and Ni in the surface dusts were regarded as carcinogenic HEs, based on the related classification list [44]. The *CR* values of these four HEs in the surface dust in all the urbanization gradients, via three exposure routes, were estimated, and then the cumulative effect of the *CR* of the analyzed HEs was estimated, using the total carcinogenic risk (*TCR*).

As given in Table 6, for both adults and children, the average *CR* of the four carcinogenic HEs in the surface dust in the core urban, urban, and suburban gradient zones can be ranked: $CR_{Cd} > CR_{As} > CR_{Ni} > CR_{Cr}$. This implies that cadmium is the main carcinogenic risk factor in surface dust, with the highest *CR* value. Meanwhile, the average values of the *CR* of the four carcinogenic HEs in the surface dust followed the order: $CR_{ingest} > CR_{dermal} > CR_{inhale}$. This indicates that the main pathway of exposure to the carcinogenic risks of the HEs in the surface dust in the study area is the ingestion route.

**Table 6.** Carcinogenic risk index of the HEs in the surface dust.

| Gradient | Metal | $CR_{ingest}$ | | $CR_{inhale}$ | | $CR_{dermal}$ | | $CR$ | | $TCR$ | |
|---|---|---|---|---|---|---|---|---|---|---|---|
| | | Children | Adults | Children | Adults | Children | Adults | Children | Adults | Children | Adults |
| Core urban | As | $1.12 \times 10^{-5}$ | $9.03 \times 10^{-6}$ | $8.87 \times 10^{-13}$ | $3.08 \times 10^{-12}$ | $3.03 \times 10^{-7}$ | $7.08 \times 10^{-7}$ | $1.15 \times 10^{-5}$ | $9.74 \times 10^{-6}$ | $3.77 \times 10^{-5}$ | $3.10 \times 10^{-5}$ |
| | Cd | / | / | $3.35 \times 10^{-11}$ | $1.16 \times 10^{-10}$ | / | / | $2.62 \times 10^{-5}$ | $2.12 \times 10^{-5}$ | | |
| | Cr | $2.61 \times 10^{-5}$ | $2.10 \times 10^{-5}$ | $6.05 \times 10^{-8}$ | $2.10 \times 10^{-7}$ | / | / | $3.35 \times 10^{-11}$ | $1.16 \times 10^{-10}$ | | |
| | Ni | / | / | $7.01 \times 10^{-10}$ | $2.43 \times 10^{-9}$ | / | / | $7.01 \times 10^{-10}$ | $2.43 \times 10^{-9}$ | | |
| Urban | As | $1.22 \times 10^{-5}$ | $9.84 \times 10^{-6}$ | $9.67 \times 10^{-13}$ | $3.36 \times 10^{-12}$ | $3.30 \times 10^{-7}$ | $7.72 \times 10^{-7}$ | $1.26 \times 10^{-5}$ | $1.06 \times 10^{-5}$ | $3.94 \times 10^{-5}$ | $3.24 \times 10^{-5}$ |
| | Cd | / | / | $2.95 \times 10^{-11}$ | $1.03 \times 10^{-10}$ | / | / | $2.69 \times 10^{-5}$ | $2.18 \times 10^{-5}$ | | |
| | Cr | $2.68 \times 10^{-5}$ | $2.16 \times 10^{-5}$ | $6.21 \times 10^{-8}$ | $2.16 \times 10^{-7}$ | / | / | $2.95 \times 10^{-11}$ | $1.03 \times 10^{-10}$ | | |
| | Ni | / | / | $6.26 \times 10^{-10}$ | $2.17 \times 10^{-9}$ | / | / | $6.26 \times 10^{-10}$ | $2.17 \times 10^{-9}$ | | |
| Suburban | As | $1.06 \times 10^{-5}$ | $8.51 \times 10^{-6}$ | $8.36 \times 10^{-13}$ | $2.91 \times 10^{-12}$ | $2.85 \times 10^{-7}$ | $6.67 \times 10^{-7}$ | $1.09 \times 10^{-5}$ | $9.18 \times 10^{-6}$ | $3.59 \times 10^{-5}$ | $2.95 \times 10^{-5}$ |
| | Cd | / | / | $2.63 \times 10^{-11}$ | $9.14 \times 10^{-11}$ | / | / | $2.51 \times 10^{-5}$ | $2.03 \times 10^{-5}$ | | |
| | Cr | $2.50 \times 10^{-5}$ | $2.01 \times 10^{-5}$ | $5.80 \times 10^{-8}$ | $2.01 \times 10^{-7}$ | / | / | $2.63 \times 10^{-11}$ | $9.14 \times 10^{-11}$ | | |
| | Ni | / | / | $5.95 \times 10^{-10}$ | $2.07 \times 10^{-9}$ | / | / | $5.95 \times 10^{-10}$ | $2.07 \times 10^{-9}$ | | |

Table 6 shows that the *TCR* values of the carcinogenic HEs in the surface dust in the core urban, urban, and suburban gradients were $3.77 \times 10^{-5}$, $3.94 \times 10^{-5}$, and $3.59 \times 10^{-5}$ for children, respectively, compared to $3.10 \times 10^{-5}$, $3.24 \times 10^{-5}$, and $2.95 \times 10^{-5}$ for adults, respectively. The calculated *TCR* values of the four carcinogenic HEs in surface dust for children were relatively higher than those for adults. This indicates that the As, Cd, Cr, and Ni elements in the surface dust may pose relatively higher carcinogenic risks to children's health.

However, according to the classification criteria for carcinogenic health risk, the *CR* or *TCR* values of the surface dust in all the urban gradients were lower than the acceptable risk threshold value ($10^{-4}$), for both children and adults. This suggests that the potential carcinogenic risks posed by the hazardous elements in surface dust are acceptable, and they do not pose a carcinogenic risk for adults or children. In addition, for both adults and children, the obtained *TCR* values of the HEs could be ranked: $TCR_{urban} > TCR_{core\ urban} > TCR_{suburban}$, indicating that the carcinogenic elements in the surface dust in the suburban gradient had a lower potential health risk than those of the other urban gradients.

The obtained results of this study showed that, regarding the *PLI* of the HEs in the surface dust along the typical urban gradient zone of Urumqi (Table 4), the *PLI* of the HEs decreased in the order: core urban > urban > suburban. Therefore, it can be reasonably concluded that the enrichment and pollution degrees of many of the analyzed HEs in the surface dust in the study area are likely to be correlated with the urbanization process. The reason is that the sample sites close to the core urban gradient are usually surrounded by dense public transport and other human activities in highly urbanized regions [45]. The core-urban–urban–suburban gradient pattern is most common in China's fast urbanization process, and it has a strong impact on the density of the traffic flow and population, as well as energy consumption [10]. Owing to the impacts of rapid developments in urbanization, the HEs in the urban surface dust may present obvious zonal distribution characteristics along a core-urban–urban–suburban gradient [46], and the concentrations of the HEs gradually decrease along urban gradient zones [47].

The results of the present study suggest that the pollution risk assessment of the HEs in surface dust could be used in the environmental quality assessment of urban areas. It is worth noting that, due to the high toxicity and health risks of Cr and Cd, these two HEs in the surface dust in the study area may pose a health risk to local residents, and to the whole ecosystem. Based on the results of the above discussions, the HEs that need priority control in all the urban gradient zones of the study area are Cr and Cd. Given the current levels of urbanization, and the fact that urbanization can lead to an increase in the HE concentration, particular attention should be paid to the levels, and the degree of health risk, of the HEs in the surface dust in the study area.

It is recommended that specific measures are taken to reduce the potential risk, and assure the protection of human health. The most effective way to alleviate HE pollution is to effectively control the pollution origins, and strictly enforce environmental regulations. A monitoring network for urban surface dust should be constructed, to ensure the systematic monitoring of the dynamic changes in HEs in the urban surface dust, to provide decision-makers with updated data on the HEs in the urban surface dust. Finally, further research work should be based on more accurate *CDI* estimation parameters, to optimize the health risk assessment of the HEs in the surface dust in arid inland cities.

## 4. Conclusions

Surface dust samples from 41 sampling sites in the core urban, urban, and suburban gradients of the city of Urumqi were collected, and the concentrations, spatial distribution, pollution levels, and potential health hazards of HEs in these samples were discussed and compared. The results indicated that:

1. The average concentrations of the Hg, Cr, Ni, and Pb elements in the surface dust in all the urbanization gradients exceeded the corresponding background values. Hg was the most enriched hazardous element in the surface dust in the study area. The spatial distributions of As and Pb were similar to one another, with higher concentrations seen in the core urban and urban gradients. Higher concentrations of Hg, Cd, and Ni were observed in the core urban gradient, whereas higher concentrations of Cr were observed in the urban gradient.

2. The average *CF* values of the Hg, Cd, and Ni in the surface dust could be ranked: core urban > urban > suburban, and the average *CF* of As, Cr, and Pb in the surface dust could be ranked: urban > core urban > suburban. The average *PLI* values of the HEs in the surface dust in the core urban, urban, and suburban gradients were 1.35, 1.29, and 1.15, respectively, with a low level of pollution. The *PLI* of the HEs could be ranked: core urban > urban > suburban.

3. The *HI* values of the HEs in the surface dust in the core urban, urban, and suburban gradients were 0.910, 0.956, and 0.839 for children, respectively, compared to 0.158, 0.166, and 0.146 for adults, respectively. Meanwhile, the *TCR* values of the carcinogenic HEs in the surface dust in the core urban, urban, and suburban gradients were $3.77 \times 10^{-5}$, $3.94 \times 10^{-5}$, and $3.59 \times 10^{-5}$ for children, respectively, compared to $3.10 \times 10^{-5}$, $3.24 \times 10^{-5}$, and $2.95 \times 10^{-5}$ for adults, respectively. The *HI* and *TCR* values of the HEs for adults and children can be ranked: urban > core urban > suburban. The potential non-carcinogenic and carcinogenic health risks of the investigated HEs, instigated primarily through the oral ingestion of surface dust, were found to be within the acceptable range. Finally, Cr was the main non-carcinogenic risk factor, while Cd was the main carcinogenic risk factor among the analyzed HEs in the surface dust in all the urbanization gradients.

**Author Contributions:** Conceptualization, N.K., M.E. and N.S.; methodology, N.K. and M.E.; software, N.K.; validation, N.K. and M.E.; formal analysis, N.K. and N.S.; investigation, M.E.; resources, N.K. and M.E.; data curation, N.K.; writing—original draft preparation, N.K.; writing—review and editing, N.K., M.E. and N.S.; visualization, M.E. and N.S.; supervision, N.K. and M.E.; project administration, M.E.; funding acquisition, M.E. All authors have read and agreed to the published version of the manuscript.

**Funding:** This research is funded by the National Natural Science Foundation of China (No. U2003301), and the Tianshan Talent Training Project of Xinjiang.

**Institutional Review Board Statement:** Not applicable.

**Informed Consent Statement:** Not applicable.

**Data Availability Statement:** Data will be available upon request to the corresponding author.

**Conflicts of Interest:** The authors declare no conflict of interest.

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
