# Peer review of "Pollution and Health Risk Assessment of Hazardous Elements in Surface Dust along an Urbanization Gradient"

_sustainability, doi:10.3390/su151511842_

Round 1
Reviewer 1 Report
Pollution and Health Risk Assessment of Hazardous Elements in Surface Dust along an Urbanization Gradient
Comments
This paper basically discussed the Pollution and Health Risk Assessment of Hazardous Elements in Surface Dust along an Urbanization Gradient. The methods and models are appropriate and well written with reasonable results and discussion but there are some concerns from my side which need to be addressed.
1. Section 1 Introduction is not interlinked please check again.
2. (Methodology Section)
3. On what specific basis you have selected the 41-sample size?
4. Why you applied these models The pollution load index (PLI) and the US EPA health risk assessment model to analyze the pollution and health risk assessments.?
5. In Section Conclusion – the discussion about the study’s implications is not satisfactory. More discussion on why and how the results from this study would have implications for other developing countries is needed. Specifically, what lessons/strategies other countries can learn from this study?
6. Lastly, the manuscript is not easy to read and there are so many grammatical errors. It must be clear before accepting.
Lastly, the manuscript is not easy to read and there are so many grammatical errors. It must be clear before accepting.
Author Response
Dear Editors and Reviewers:Thank you for your letter and the opportunity to revise our paper on “Pollution and Health Risk Assessment of Hazardous Elements in Surface dust along an Urbanization Gradient” (sustainability-2492279). We owe sincere thanks to the reviewers for their time spent on reviewing our manuscript and the suggestions offered by the reviewers have been immensely helpful. Those comments and suggestions are all valuable and very helpful for revising and improving our paper, as well as the important guiding significance to our researches. We have studied comments carefully and have made correction which we hope meet with approval. Based on your comment and request, we have made extensive modification on the original manuscript. Revised portion are marked in blue on the revised manuscript. The main corrections in the paper and the responds to the reviewer’s comments are as flowing:
Responds to the reviewer’s comments: (The responses are in red, and revised portions are marked blue in the letter.)
Reviewer 1: Comments and Suggestions for Authors This paper basically discussed the Pollution and Health Risk Assessment of Hazardous Elements in Surface Dust along an Urbanization Gradient. The methods and models are appropriate and well written with reasonable results and discussion but there are some concerns from my side which need to be addressed.1. Section 1 Introduction is not interlinked please check again.
Authors: Thanks for your favorable suggestions. We have made corrections on that. (From Line 53 to Line 73 of the revised manuscript)
2. (Methodology Section)a. On what specific basis you have selected the 41-sample size?
Authors: We have added the sample collection methods used in this study. (From Line 105 to Line 106 of the revised manuscript)
b. Why you applied these models the pollution load index (PLI) and the US EPA health risk assessment model to analyze the pollution and health risk assessments?
Authors: Thanks for your good comments. We have improved the Materials and Methods part of the manuscript according to your suggestion. (From Line 122 to Line 124, and from Line 134 to Line 136 of the revised manuscript)
3. In Section Conclusion – the discussion about the study’s implications is not satisfactory. More discussion on why and how the results from this study would have implications for other developing countries is needed. Specifically, what lessons/strategies other countries can learn from this study?
Authors: According to your advice, we have rewritten the discussion. (From Line 333 to Line 357 of the revised manuscript)
4. Lastly, the manuscript is not easy to read and there are so many grammatical errors. It must be clear before accepting.
Authors: Many thanks for your suggestions. We are very sorry for our incorrect writing. According with your advice, we have made corrections on that. (All parts of the revised manuscript)
We tried our best to improve the manuscript. We appreciate for your warm work earnestly, and hope that the correction will meet with approval.
Once again, many thanks for your dedicated work.
Sincerely,
Mamattursun EZIZ
2023/19/07

Reviewer 2 Report
This was an interesting paper that demonstrated or attempted to demonstrate the chemical and physical make up of surface dust. The methods and results appear to be appropriate (although the analysis could be made clearer) There is limited information on the specific point of the manuscript - the title would suggest that there would be a way of interpreting the results in relation to the health outcomes, however the only point that the impact on health is discussed in the the discussion. Does the make up of the sand make it itself more hazardous, or is this all conjecture based on a model? Further to this - the discussion needs re-writing with more information and details discussion on the relevance of the results - at present it just describes them again.I feel the grammar was readable but it would benefit from an English language review
Author Response
Dear Editors and Reviewers:Thank you for your letter and the opportunity to revise our paper on “Pollution and Health Risk Assessment of Hazardous Elements in Surface dust along an Urbanization Gradient” (sustainability-2492279). We also owe sincere thanks to the reviewers for their time spent on reviewing our manuscript and the suggestions offered by the reviewers have been immensely helpful. Those comments and suggestions are all valuable and very helpful for revising and improving our paper, as well as the important guiding significance to our researches. We have studied comments carefully and have made correction which we hope meet with approval. Based on your comment and request, we have made extensive modification on the original manuscript. Revised portion are marked in blue on the revised manuscript. The main corrections in the paper and the responds to the reviewer’s comments are as flowing:
Responds to the reviewer’s comments: (The responses are in red, and revised portions are marked blue in the letter.)
Reviewer 2:
Comments and Suggestions for Authors This was an interesting paper that demonstrated or attempted to demonstrate the chemical and physical make up of surface dust. The methods and results appear to be appropriate (although the analysis could be made clearer). There is limited information on the specific point of the manuscript - the title would suggest that there would be a way of interpreting the results in relation to the health outcomes, however the only point that the impact on health is discussed in the discussion. Does the make up of the sand make it itself more hazardous, or is this all conjecture based on a model? Further to this - the discussion needs re-writing with more information and details discussion on the relevance of the results - at present it just describes them again.
Authors: Thanks for your favorable suggestions. According with your advice, we have rewritten the discussion. (From Line 333 to Line 357 of the revised manuscript)
We tried our best to improve the manuscript. We appreciate for your warm work earnestly, and hope that the correction will meet with approval.
Once again, many thanks for your dedicated work.
Sincerely,
Mamattursun EZIZ
2023/19/07

Reviewer 3 Report
The authors presented an important assessment of hazardous elements in surface dust and the potential health risk for adults and children. However, several points may be better discussed; please, find attached the specific comments. In addition, English must be reviewed.
Lines 53-83: this section must be entirely rewritten: the authors can’t report other studies such as a list. In addition, a language revision is strongly recommended.
Lines 85-86: several studies also focused on dust metal pollution in indoor environments, such as houses, schools, and museums. Please add some references to the text.
Line 89: see the first comment (lines 53-83).
Line 105: “arid zone oasis” seems not correct. What do the authors mean?
Line 114: the authors could specify how the distinction between core urban-urban-suburban gradient was identified: is it based on population density or similar features?
Line 116: What is the meaning of “extended over a distance of about 8 km”? It is a radius or a surface extension?
Line 124: How the authors could establish the heterogeneity of HEs in surface dust before the start of the experiment?
Line 148: How the background concentration of each element was calculated? Please, specify in the text.
Line 149: the pollution grades were decided by the authors or are based on previous papers?
Line 158: Why the relative bioavailability following ingestion was not considered? This is probably the most important parameter to consider to perform a risk analysis, as suggested by the US EPA Exposure Handbook.
Line 172: explain what is the meaning of Reference dose and report where the values of Table 2 have been found.
Line 176: was the TCR calculated for all the HEs of the dust surface? Please explain.
Lines 213-215: Hg could be released by other anthropogenic sources. Please, report other examples.
Line 238: the verb tense of the sentence seems not correct.
Lines 361-362: Why do the authors suppose this?
Line 372: correct the sentence.
I suggest that all the manuscript, and in particular the introduction section, must be reviewed.
Author Response
Dear Editors and Reviewers:Thank you for your letter and the opportunity to revise our paper on “Pollution and Health Risk Assessment of Hazardous Elements in Surface dust along an Urbanization Gradient” (sustainability-2492279). We also owe sincere thanks to the reviewers for their time spent on reviewing our manuscript and the suggestions offered by the reviewers have been immensely helpful. Those comments and suggestions are all valuable and very helpful for revising and improving our paper, as well as the important guiding significance to our researches. We have studied comments carefully and have made correction which we hope meet with approval. Based on your comment and request, we have made extensive modification on the original manuscript. Revised portion are marked in blue on the revised manuscript. The main corrections in the paper and the responds to the reviewer’s comments are as flowing:
Responds to the reviewer’s comments: (The responses are in red, and revised portions are marked blue in the letter.)
Reviewer 3:
Comments and Suggestions for Authors
The authors presented an important assessment of hazardous elements in surface dust and the potential health risk for adults and children. However, several points may be better discussed; please, find attached the specific comments. In addition, English must be reviewed.1. Lines 53-83: this section must be entirely rewritten: the authors can’t report other studies such as a list. In addition, a language revision is strongly recommended.DoneAuthors: Many thanks for your favorable suggestions. We are very sorry for our incorrect writing. We have rewritten the Introduction part. (From Line 54 to Line 73 of the revised manuscript) 2. Lines 85-86: several studies also focused on dust metal pollution in indoor environments, such as houses, schools, and museums. Please add some references to the text.Authors: Thanks for your suggestions. We have added related researches. (From Line 66 to Line 67 of the revised manuscript) 3. Line 89: see the first comment (lines 53-83).Authors: We have made corrections on that. (From Line 68 to Line 70 of the revised manuscript) 4. Line 105: “arid zone oasis” seems not correct. What do the authors mean?Authors: We have corrections on that. (Line 80 of the revised manuscript) 5. Line 114: the authors could specify how the distinction between core urban-urban-suburban gradient was identified: is it based on population density or similar features?Authors: Thanks for your good comments. According with your advice, we have made corrections on that. (From Line 91 to Line 95 of the revised manuscript) 6. Line 116: What is the meaning of “extended over a distance of about 8 km”? It is a radius or a surface extension?Authors: We are very sorry for our mistakes. We have deleted that. 7. Line 124: How the authors could establish the heterogeneity of HEs in surface dust before the start of the experiment?Authors: We have added the related references for that. (From Line 102 to Line 103 of the revised manuscript) 8. Line 148: How the background concentration of each element was calculated? Please, specify in the text.Authors: We have added the related references for that. (Line 128 of the revised manuscript) 9. Line 149: the pollution grades were decided by the authors or are based on previous papers?Authors: Thanks for your good comments. We have added the related references for that. (From Line 131 to Line 132 of the revised manuscript) 10. Line 158: Why the relative bioavailability following ingestion was not considered? This is probably the most important parameter to consider to perform a risk analysis, as suggested by the US EPA Exposure Handbook.Authors: Thanks for your favorable suggestions. The methods used in our study has been widely used to evaluate the pollution status HEs in surface dusts. (As described in Jiang et al (2018), Adila et al (2020), Wang et al (2021) , Wang et al (2022), Shahab et al (2022)… in the references list of this manuscript) 11. Line 172: explain what is the meaning of Reference dose and report where the values of Table 2 have been found.Authors: Thanks for your favorable suggestions. We have made corrections on that. (From Line 154 to Line 155, and from Line 168 to Line 169 of the revised manuscript) 12. Line 176: was the TCR calculated for all the HEs of the dust surface? Please explain.Authors: Thanks for your favorable suggestions. We have explained it from Line 163 to Line 164 of the revised manuscript. 13. Lines 213-215: Hg could be released by other anthropogenic sources. Please, report other examples.Authors: According to your advice, we have made corrections on that. (From Line 198 to Line 199 of the revised manuscript) 14. Line 238: the verb tense of the sentence seems not correct.Authors: We have made corrections on that. (From Line 223 to Line 235 of the revised manuscript) 15. Lines 361-362: Why do the authors suppose this?Authors: We are very sorry for our incorrect writing. According with your advice, we have made corrections on that. (From Line 354 to Line 363 of the revised manuscript) 16. Line 372: correct the sentence.Authors: Thanks for your good comments. We have made corrections on that. (From Line 370 to Line 372 of the revised manuscript)
We tried our best to improve the manuscript. We appreciate for your warm work earnestly, and hope that the correction will meet with approval.
Once again, many thanks for your dedicated work.
Sincerely,
Mamattursun EZIZ
2023/19/07

Round 2
Reviewer 1 Report
The author well defines the comments. I am satisfied.